# OpenReview forum: "Monitorability as a Free Gift: How RLVR Spontaneously Aligns Reasoning"
_ICML.cc/2026/Conference — ICML 2026 regular_

### Official Review · Reviewer_iMp4 · 2026-03-01

**Soundness:** 4
**Presentation:** 4
**Significance:** 3
**Originality:** 3
**Overall Recommendation:** 5
**Confidence:** 4

**Summary:**

This work investigated the monitorability of LRMs trained using RLVR. The main metric used was the g-mean2. RLVR improves monitorability in ways that generalize across domains, but gains depend strongly on training data, with instruction-following and diverse datasets helping most. Monitorability is distinct from capability and is associated with lower-entropy, more deterministic responses and attention shifts toward the prompt. Training on medium-difficulty tasks is most effective, while overly hard tasks yield limited improvements.

**Compliance With Llm Reviewing Policy:**

Affirmed.

**Key Questions For Authors:**

- Do you have any experiments or guesses on what happens for agentic domains or alignment training?
- Do you think monitorability can be optimized as a reward? like using the g-mean2 as reward. Will it impact accuracy? maybe it will optimize for entropy?

**Limitations:**

negative societal impact was discussed but missing standalone limitations section

**Strengths And Weaknesses:**

# Strengths
- very comprehensive experiments, also investgated cross domain results
- interesting findings that confirm but also some opposed to the conventional understanding of monitorability
- discovery of IF training significantly boosts monitorability is actionable finding for people who care to maximize it
- other cool results such as attention / entropy minimization

# Weaknesses
- missing mid sized models, size generalizable claims couldnt be made
- different model classes, such as llama missing, where capability could still be required to have decent monitorability, more like a emergent capability?

---

> ### Author Rebuttal · Authors · 2026-03-31
>
> We thank the reviewer for the positive feedback and recognition of our contributions, especially the findings on IF training, cross-domain behavior, and mechanistic analysis. We address weakness and question as follows:
>
>
>
> ---
>
> **Weakness: Size generalizability**
>
> To address generalization across model scale and architecture, we conduct additional experiments on a larger model from a different family (**DeepSeek-R1-Distill-Llama-8B**):
>
> | DS-8B |  Step100 | Step200 | Step300 | Step400 |
> | ----- | ---- | ------- | ------- | ------- |
> | Math  | 0.40 | 0.43    | 0.42    | 0.42    |
> | IF    | 0.42 | 0.47    | 0.50    | 0.48    |
>
> We observe consistent patterns:
>
> * **early-stage improvement**, and
> * **IF training improves monitorability**.
>
> To further examine the relationship with capability, we compute the agreement ratio of step-wise changes: $\mathbb{1}[\Delta \text{Cap} \cdot \Delta \text{GMean}^2 > 0],$
> which is **0.6** across datasets, with 0.5 as random guess. This supports our claim that better capability does not often yield better monitorability, even at larger scales.
>
>
> ---
> **Q1: What happens in agentic domains or alignment training?**
>
> Thank you for this insightful question. While we do not directly study agentic training, our results suggest the following hypotheses:
>
> * Agentic domains (e.g., search agents):
>   These often require **structured action formats and grounded reasoning before execution**, similar to IF training. This may improve monitorability by enforcing stronger coupling between prompt, reasoning, and action.
>   However, agentic settings also favor **shorter, efficiency-driven reasoning**, which may reduce observable traces and thus hurt monitorability, as discussed in [1].
>
> * Alignment training (e.g., deliberative alignment, confession-style training):
>   These methods explicitly encourage **reflection, justification, and sensitivity to prompt nuances**, which align closely with the properties in Instruction Following training as discussed in **Response W1 of Reviewer ESWC**
> . Therefore, they are likely to improve monitorability.
>
> ---
>
> **Q2: Can monitorability be optimized as a reward?**
>
> Thank you for this important question. We believe directly optimizing monitorability metrics (e.g., g-mean2) as a reward is always risky, as it may lead to reward hacking, especially when the metric depends on an external LLM monitor, as shown in [1].
>
> However, if such reward apply, combining monitorability-related signals with verifiable task rewards (e.g., math, code) may still help preserve accuracy.
>
> We also study the role of entropy via controlled experiments (higher-clip GRPO):
>
> **Math/IF entropy control (DS-1.5B)**
>
> | Setup            | 100         | 200         | 300         | 400         | max (IF/Math)   |
> | ---------------- | ----------- | ----------- | ----------- | ----------- | ----------- |
> | IF               | 0.30 (0.85) | 0.36 (0.80) | 0.38 (0.75) | 0.38 (0.74) | 0.39 (0.71) |
> | IF (Clip High)   | 0.31 (0.87) | 0.35 (0.83) | 0.40 (0.81) | 0.38 (0.81) | -           |
> | Math             | 0.29 (0.84) | 0.33 (0.80) | 0.34 (0.79) | 0.36 (0.78) | 0.36 (0.78) |
> | Math (Clip High) | 0.32 (0.86) | 0.35 (0.87) | -           | -           | -           |
>
> For max (IF/Math 800), we report the maximum monitorability within 800 steps for IF and Math. Due to time limits, we did not finish running over 800 steps for both experiments. However, preliminary results suggest that controlling entropy (i.e., encouraging exploration and delaying collapse) can potentially improve monitorability under the same training data (better monitorability for IF at 300 steps, and higher step-wise monitorability for Math.) This supports the view that training dynamics, rather than explicit reward shaping on monitorability, are a safer and more effective lever.
>
> ---
> [1] Monitoring Monitorability

---

> > ### Author Rebuttal · Reviewer_iMp4 · 2026-04-01
> >
> > Added additional experiments and gave reasonable hypothesis

---

> > > ### Author Response · Authors · 2026-04-01
> > >
> > > We thank Reviewer iMp4 again for the positive feedback and recognition of our contributions! We are glad that our follow-up has addressed your question.

---

### Official Review · Reviewer_ESWC · 2026-03-11

**Soundness:** 2
**Presentation:** 2
**Significance:** 2
**Originality:** 2
**Overall Recommendation:** 2
**Confidence:** 4

**Summary:**

This paper presents an empirical study on the dynamics of monitorability in LRMs trained with RLVR. The authors investigate whether the commonly observed improvement in monitorability during early RLVR training is a universal phenomenon. The paper analyzes how training data distribution, model capability, and task difficulty influence monitorability. The results show that monitorability improvements are dependent on the training data distribution, particularly benefiting from instruction-following data and diverse datasets. The paper further finds that monitorability is largely orthogonal to task capability, and mechanistic analyses suggest that improvements in monitorability are mainly associated with entropy reduction (distribution sharpening) and increased attention to the prompt rather than stronger causal reliance on reasoning traces. The authors also validate their findings with additional metrics such as Draft-to-Answer faithfulness and examine the role of training and evaluation difficulty.

**Compliance With Llm Reviewing Policy:**

Affirmed.

**Final Justification:**

My main concerns remain only partially addressed. In particular, the rebuttal still does not clearly identify the underlying causal factors behind the effect of IF data on monitorability, most of the new evidence remains correlational. Similarly, while the entropy analysis is refined, it does not fully resolve the inconsistencies or provide a concrete mechanistic explanation.

Overall, the paper is strengthened empirically, but the core issues regarding causal understanding and explanatory depth remain. Therefore, my overall assessment remains unchanged.

**Key Questions For Authors:**

Some experimental design choices lack sufficient justification. For example, in Line 182, the authors filter out cases where behavior changes are below 5%. It is unclear why this specific threshold is chosen. Is this a standard practice in prior work, or is there an empirical rationale behind it? Additional justification would strengthen the experimental methodology.

**Strengths And Weaknesses:**

### Strengths

* The paper presents an extensive empirical study of monitorability. It evaluates two base models under multiple training data recipes and across a wide range of benchmarks and evaluation metrics. This comprehensive experimental setup provides a clearer empirical characterization of monitorability.

### Weaknesses

While the paper presents many experiments and reports several interesting observations, some conclusions lack sufficiently clear and in-depth analysis. In several cases, the explanations remain at the level of hypotheses rather than verified mechanisms, which limits both the depth of the paper and its practical implications. Specifically:

* Around Line 211, the paper claims that monitorability depends strongly on the training data distribution and that instruction-following (IF) data can significantly improve monitorability. Later, around Line 307, the authors further argue that IF capability itself is not correlated with monitorability. While I understand the distinction the authors are making, a fundamental question remains unanswered: what specific properties of IF data actually lead to improved monitorability? The paper explores many training data recipes but does not identify the underlying causal factors. Clarifying this point would be highly valuable for guiding practical RL training.

* In Figure 5a (Section 5.1), I notice that the correlation between entropy and monitorability is weak on several benchmarks. However, the paper mainly discusses cases where the correlation is strong and largely ignores the weaker or inconsistent results. As a result, the presented conclusion may overstate the relationship between entropy reduction and monitorability.

* Around Line 434, the authors hypothesize that training on hard problems leads to sparser reward signals, which in turn prevents sufficient distribution sharpening and results in lower monitorability. However, this hypothesis appears testable—for example, by directly measuring entropy as in Section 5.1. Instead of validating this mechanism empirically, the paper leaves it as a speculation.

Overall, the paper reads more like a collection of empirical observations. While the coverage is broad, the analysis lacks deeper synthesis or theoretical insight that could explain the observed phenomena.

---

> ### Author Rebuttal · Authors · 2026-03-31
>
> We thank the reviewer for their thoughtful assessment of our work. We address the weaknesses and question below:
>
> ---
>
> **W1: What causal factors of IF improve monitorability?**
>
> We agree that identifying the underlying driver is critical. Our evidence suggests the gain does not come from IF capability itself, but from a **shift in training objective**: instruction-following RL rewards **fine-grained conditioning on prompt constraints** and often **strict answer formatting grounded in reasoning**
>
> To probe these, we analyze three quantities:
>
> 1. **Reasoning/Answer-to-Prompt attention mass** (Sec. 4),
> 2. **Valid intervention ratio** (% of examples with Total Effect (>0), Line 721), which measures how often prompt interventions produce detectable answer changes.
> 3. **Hint causality ratio** (% of Positive Monitor Judgment (Z) = 1, Line 677), which measures how often the intervention meaningfully influences reasoning and thus affect answers.
>
> **Reasoning-to-Prompt attention mass**
>
> |Model| Non-IF |W/IF |
> |--|--|--|
> |DS-1.5B  |  0.112 | **0.122** |
> |Qwen3-4B |  0.110 | **0.115** |
>
> **Answer-to-Prompt attention mass**
>
> |Model| Non-IF |W/IF |
> |--|--| --|
> |DS-1.5B  | 0.087 | **0.133** |
> |Qwen3-4B | 0.068 | **0.076** |
>
> **Valid intervention ratio (%)**
>
> |Model | Non-IF | W/IF |
> |--| -- | -- |
> |DS-1.5B  | 54.1 | **58.9** |
> |Qwen3-4B | 52.7 | **58.9** |
>
> **Hint causality ratio (%)**
>
> |Model| Non-IF | W/IF |
> |--| --| -- |
> |DS-1.5B  | 21.3 | **28.7** |
> |Qwen3-4B | 41.3 | **45.7** |
>
> These consistently show that IF training:
> * increases **cross-segment attention to the prompt** (especially Answer-to-Prompt),
> * increases **answer sensitivity to prompt interventions**, and
> * increases **explicit causal reflection of interventions in reasoning and answering.**
>
> Together, this supports that IF strengthens **prompt-conditioned reasoning** and **tight casulity of reasoning and answering** making behavior more monitorable. We agree this is not fully causal, and will clarify this limitation.
>
> ---
>
> **W2/W3: Entropy–monitorability relationship**
>
> We agree the correlation is **not uniformly strong**. Our intended claim is that monitorability improves when training sharpens the policy toward **valid reasoning patterns**, but this depends on data. For example, with measurment over
> **DS-1.5B Math hardness - GMean^2 (entropy)**
>
> |Hardness  | 100 | 200 | 300 | 400 |
> |--|--|--|--|--|
> |Easy | 0.30 (0.85) | 0.34 (0.82) | 0.37 (0.79) | 0.39 (0.77) |
> |Medium | 0.29 (0.84) | 0.33 (0.80) | 0.34 (0.79) | 0.36 (0.78) |
> |Hard | 0.20 (0.79) | 0.21 (0.76) | 0.20 (0.73) | 0.21 (0.72) |
>
> Hard problems with sparse reward show **faster entropy collapse but worse monitorability**, indicating that entropy alone is insufficient—early collapse toward suboptimal distribution hurts. We acknowledge that this result **refines our original hypothesis**, and we will revise the paper accordingly.
>
> To quantify the dynamics more directly of reported entropy results, we track the fraction of training transitions between steps where entropy and monitorability move in opposite directions, namely:
> $\mathbf{1}[\{\Delta \text{Ent} \cdot \Delta \text{GMean}^2 < 0\}]$
>
> **Opposite entropy–monitorability ratio (%)**
>
> |Benchmark | DS-1.5B | Qwen3-4B |
> | --| -- | --|
> |AMC12|56.5 |64.3 |
> |CodeMMLU  |61.7 | 60.9 |
> |GPQA|76.6 | 65.2 |
> |MedQA|68.1 |67.4 |
>
> While generally above random (50%), we observe outliers that show degraded alignment. For example, DS-1.5B with IF only achieves 16.7% on AMC12 and Qwen3-4B Science achieves 28.6% on CodeMMLU and GPQA. These results suggest that collapsing into a suboptimal mode can worsen monitorability.
>
> We thus further control entropy via higher-clip ($\epsilon=0.28$) GRPO [1] on DS1.5B
>
> **Math/IF entropy control (DS-1.5B): monitorability (entropy)**
>
> |Setup | 100 | 200 | 300| 400| max (IF/Math 800)   |
> |-- | --| -- | --| --| -- |
> |IF | 0.30 (0.85) | 0.36 (0.80) | 0.38 (0.75) | 0.38 (0.74) | 0.39 (0.71) |
> |IF (Clip High)| 0.31 (0.87) | 0.35 (0.83) | 0.40 (0.81) | 0.38 (0.81) | - |
> |Math | 0.29 (0.84) | 0.33 (0.80) | 0.34 (0.79) | 0.36 (0.78) | 0.36 (0.78) |
> |Math (Clip High) | 0.32 (0.86) | 0.35 (0.87) | -  | - | - |
>
> For max (IF/Math 800), we report the maximum monitorability within 800 steps for IF and Math. Due to time limits, we did not finish all running. However, preliminary results suggest that controlling entropy (i.e., encouraging exploration and delaying suboptimal collapse) can potentially improve monitorability under the same training data (better monitorability for IF at 300 steps, and higher step-wise monitorability for Math.)
>
> For other RL guidelines, we refer to Response **W3/Q3 to Reviewer 476P**
>
> ---
>
> **Q1: 5% filter**
>
> The adoption of filter follows prior work [1] and 5% ratio is empirically motivated. With ~200 examples and 3 interventions, this ensures at least ~30 valid cases and make estimation more stable.
>
> ---
> [1] DAPO: an Open-Source LLM Reinforcement Learning System at Scale
>
> [2] Monitoring Monitorability

---

> > ### Author Rebuttal · Reviewer_ESWC · 2026-04-03
> >
> > Thank you for the detailed rebuttal and additional analyses. I appreciate the effort to provide further empirical evidence and clarify some claims.
> >
> > However, my main concerns remain only partially addressed. In particular, the rebuttal still does not clearly identify the underlying causal factors behind the effect of IF data on monitorability, most of the new evidence remains correlational. Similarly, while the entropy analysis is refined, it does not fully resolve the inconsistencies or provide a concrete mechanistic explanation.
> >
> > Overall, the paper is strengthened empirically, but the core issues regarding causal understanding and explanatory depth remain. Therefore, my overall assessment remains unchanged.

---

> > > ### Author Response · Authors · 2026-04-06
> > >
> > > Thank you again for the careful follow-up. We agree that strengthening the causal understanding is central to improving this work, and we appreciate the reviewer’s emphasis on this point.
> > >
> > > For the **entropy mechanism**, we have now completed the 800-step higher-clip GRPO runs and added a more controlled experiment: after **400 steps of higher-clip GRPO** ($\epsilon_{high}=0.28$), we continue training for **400 additional steps with standard GRPO clipping** ($\epsilon_{high}=0.2$). This allows us to separate two phases:
> > >
> > > * an initial **exploration phase** with delayed entropy collapse and encourage explorations with high reward, and
> > > * a later **sharpening phase** on the same training data.
> > >
> > > This isolates the effect of **entropy reduction after exploration**, without changing data or reward.
> > >
> > > **Math/IF Entropy Control (DS-1.5B) -- GMean^2(Entropy)**
> > >
> > > | Model| 100| 200| 300  | 400  | 500 | 600| 700 | 800 |
> > > | -- | --| --| -- | -- | --| -- | -- | --- |
> > > | IF   | 0.30 (0.85) | 0.36 (0.80) | 0.38 (0.75) | 0.38 (0.74) | 0.38 (0.71) | 0.38 (0.70) | 0.36 (0.70) | 0.39 (0.71) |
> > > | IF (Clip High)   | 0.31 (0.87) | 0.35 (0.83) | 0.40 (0.81) | 0.38 (0.81) | 0.38 (0.80) | 0.37 (0.79) | 0.34 (0.80) | 0.33 (0.80) |
> > > | Math| 0.29 (0.84) | 0.33 (0.80) | 0.34 (0.79) | 0.36 (0.78) | 0.34 (0.76) | 0.36 (0.74) | 0.35 (0.73) | 0.36 (0.71) |
> > > | Math (Clip High) | 0.32 (0.86) | 0.35 (0.87) | 0.35 (0.87) | 0.36 (0.87) | 0.34 (0.87) | 0.34 (0.87) | 0.34 (0.87) | 0.34 (0.87) |
> > >
> > > These runs show that **maintaining high entropy alone does not continue to improve monitorability**. Instead, monitorability becomes roughly flat (Math) or even declines (IF) during prolonged exploration. Along with the trend observed in standard RLVR setups, we confirm that the gain in monitorability mainly comes from **initial entropy reduction and distribution sharpening**, and then remains steady or even declines once the entropy stabilizes.
> > >
> > > We then test whether **subsequent sharpening from a better-explored state** improves monitorability under the **same data**:
> > >
> > > **Controlled collapse after exploration**
> > >
> > > | Model  | 500 | 600 | 700 | 800  |
> > > | -- | --  | --| -- | -- |
> > > | IF (0.28)   | 0.38 (0.80) | 0.37 (0.79) | 0.34 (0.80) | 0.33 (0.80) |
> > > | IF (0.28->0.2)   | 0.38 (0.76) | 0.39 (0.73) | 0.39 (0.71) | 0.37 (0.70) |
> > > | Math (0.28)  | 0.34 (0.87) | 0.34 (0.87) | 0.34 (0.87) | 0.34 (0.87) |
> > > | Math (0.28->0.2) | 0.35 (0.84) | 0.38 (0.81) | 0.38 (0.79) | 0.38 (0.77) |
> > >
> > > Where the number in brackets is $\epsilon_{high}$. Here we observe a clear improvement once training switches back to standard clipping and allows entropy to decrease from the better-explored regime. Since the **training data is unchanged**, this provides stronger support that the **trajectory of entropy collapse itself** plays a causal role. More precisely, our interpretation is:
> > > * **Premature collapse** can hurt monitorability by locking the policy into suboptimal reasoning modes.
> > > * **Sustained high entropy** alone does not improve and can hurt monitorability.
> > > * **Sharpening after sufficient exploration** improves or maintains monitorability, suggesting that what matters is not entropy reduction in isolation, but entropy reduction toward **valid reasoning modes**.
> > >
> > > For the **IF effect**, we also agree that the rebuttal evidence remains correlational. At the same time, we believe it meaningfully narrows the hypothesis. Specifically, IF training consistently increases:
> > >
> > > * **Answer-to-Prompt attention mass**,
> > > * **valid intervention ratio**, and
> > > * **hint causality ratio**.
> > >
> > > Together, these results suggest a more specific interpretation: IF training improves measured monitorability by encouraging tighter conditioning of reasoning and answers on prompt constraints, increasing the likelihood that interventions propagate to final behavior. While not a definitive causal identification, this makes the explanation more concrete and testable.
> > >
> > > More broadly, we want to clarify our contribution. We agree that a full causal account of monitorability remains open, but the paper still contributes critically to the underexplored monitorability of reasoning models:
> > >
> > > * the first detailed **dynamic-level study** of how monitorability evolves during RLVR,
> > > * a clear separation of monitorability from **training/testing distribution**, **raw capability**, and **hardness of reasoning**,
> > > * **practical training recipes** (e.g., mixed or staged IF training, exploration, then collapse helps),
> > > * controlled evidence that **entropy dynamics matter** (including the clip-control results), and
> > > * identification of failure modes in common monitorability metrics, especially due to prompt sensitivity and attention shifts.
> > >
> > > We acknowledge that the causal analysis is still incomplete and will state this more clearly in the revision. We also hope the reviewer will reconsider the overall assessment in light of the additional controlled entropy results and the paper’s contributions.

---

### Official Review · Reviewer_xozf · 2026-03-13

**Soundness:** 2
**Presentation:** 3
**Significance:** 3
**Originality:** 2
**Overall Recommendation:** 4
**Confidence:** 4

**Summary:**

This paper studies when RLVR makes a model’s chain-of-thought more monitorable, meaning easier for an external monitor to tell when the model was influenced by misleading hints or unsafe instructions. The authors train two small reasoning models, Qwen3-4B and DeepSeek-R1-Distill-Qwen-1.5B, with GRPO on different RLVR data mixtures: math, code, science, instruction following (IF), mixed data, and a cascade that adds IF later. They evaluate monitorability mainly with g-mean^2 under hint interventions, and they also compare against capability benchmarks and a secondary faithfulness measure, Draft-to-Answer (D2A).

**Compliance With Llm Reviewing Policy:**

Affirmed.

**Final Justification:**

The additional experiments and clarifications strengthened the novelty, so I recommend weak acceptance.

**Key Questions For Authors:**

1. Can you rerun the IF / ALL / w/o IF / +IF comparisons with matched total RL budget? Right now, +IF appears to receive a longer optimization schedule than the other conditions. If the IF advantage remains after matching steps and total samples seen, my confidence in the paper’s main practical claim would increase a lot.
2. How stable are the main trends across random seeds, bootstrap confidence intervals, and different monitor prompts/models? The paper reports one alternate monitor correlation, which is helpful, but I would like uncertainty estimates for the main curves and checkpoint correlations. Strong stability evidence would improve my soundness score.
3. Can you provide a more causal test of the entropy/attention explanation? For example, control entropy directly, add entropy regularization, or intervene on prompt/reasoning segments during analysis. If those interventions reproduce the claimed effects, the mechanism section becomes much stronger.
4. How much of g-mean² is really “monitorability” versus prompt-following? Your prompt-attention finding suggests a possible metric confound. If you can show stronger agreement with non-prompt-based faithfulness tests, I would trust the main conclusion more.
5. Do the same conclusions hold for larger models or models without prior long-CoT cold start? If yes, the paper’s significance would increase substantially; if not, the paper should narrow its claims more explicitly.

**Limitations:**

No. The paper mentions very general societal misuse, but it should discuss methodological limitations more directly.

**Strengths And Weaknesses:**

Strengths:
1. The paper asks an important and timely safety question: whether standard RLVR makes reasoning traces easier or harder to audit. That is directly relevant to current reasoning-model development.
2. The empirical scope is broad. The study covers multiple training domains, two model families, many evaluation domains, monitorability metrics, capability metrics, and extra settings like sycophancy, sandbagging, and task difficulty.
3. The result that instruction-following data appears especially helpful for monitorability is practically interesting and potentially actionable for future RL pipelines. The claim that monitorability is partly separate from raw capability is also useful.
4. I appreciate that the paper does not fully trust its own main metric. The authors notice that prompt-directed attention may create a “false sense of monitorability,” and they add D2A as a second check. That makes the paper more careful than a pure single-metric study.

Weaknesses:
1. Novelty is limited relative to the closest prior work. The paper does not introduce the main “free gift” observation, nor the main monitorability metric, nor the idea that faithfulness is fragile. Those pieces already appear in [1][2][3]. The main new part is a broader empirical sweep over data mixtures and difficulty.
2. Some of the strongest empirical claims are confounded by training budget and setup. The “+IF” cascade gets 600 steps of mixed training plus 600 steps of IF fine-tuning, while the other settings are described as 800-step runs. That makes the claim that IF is a late-stage “monitorability booster” less clean than it should be. The data-mixture comparisons also change more than one thing at once, including domain mix and likely reward density.
3. The main metric may partly capture prompt sensitivity or instruction-following, not true reasoning faithfulness. The authors themselves find that monitorability correlates more with attention to the prompt than with answer-to-reasoning attention, and D2A only partially resolves this concern because it is run on a subset of tasks and is mixed for some domains on DS-1.5B.
4. The “mechanistic” claims are mostly correlational, and the generality is limited. Entropy and attention analyses are interesting, but they do not show causation. Also, the experiments use only two relatively small models that already had long-CoT cold starts, so it is unclear how far the conclusions transfer to larger frontier systems. That caution matters especially because [3] studies frontier-scale monitorability and report somewhat different optimization behavior.

[1] Reasoning Models Don't Always Say What They Think
[2] Measuring the Faithfulness of Thinking Drafts in Large Reasoning Models
[3] Monitoring Monitorability

---

> ### Author Rebuttal · Authors · 2026-03-31
>
> We thank the reviewer for the careful review. We address the weaknesses and questions as follows.
>
> ---
> **W1: Novelty is limited relative to prior work**
>
> We agree that our paper does not introduce the original “free gift” observation or the monitorability metric. However, our contribution is **not another empirical confirmation**, but a **systematic, training-dynamics analysis of how monitorability emerges during RLVR**, which is missing in prior work.
>
> More specifically, our work provides the first **dynamic, checkpoint-level analysis**, leading to several findings that cannot be obtained from static evaluations.
>
> ---
> **W2/Q1: Controlled training budget**
>
> Thank you for pointing out this important confound. We have rerun experiments with matched budgets.
> We start from a 400-step checkpoint trained without IF, and then apply **+400 steps of IF tuning**, ensuring equal total updates:
>
> |Model| IF | ALL | w/o IF | +IF (400+400) |
> | -- | -- | --| -- | -- |
> | **DS-1.5B** | 0.388±0.018 | 0.500±0.016 | 0.420±0.019 | 0.478±0.016 |
> | **Qwen3-4B** | 0.603±0.022 | 0.633±0.018 | 0.543±0.025 | 0.626±0.021 |
>
> Under matched budgets, **+IF still yields a clear improvement over w/o IF**, confirming that the effect is not due to longer optimization.
>
> For data-mixture experiments, we isolate **with/without IF** as the primary variable, motivated by the observed strong and consistent IF effect on the single domain.
>
> ---
> **W3 / Q4: Metric may not reflect true reasoning faithfulness**
>
> We agree this is an important concern. However, our position is that this is not a limitation of our work, but a key finding of the paper.
> Specifically:
> * Prompt-based intervention metrics (used in [1][3]) are widely adopted.
> * Our results show that these metrics can overestimate monitorability,
> * The partial mismatch between g-mean^2 and D2A demonstrates that “monitorability” can reflect prompt sensitivity rather than reasoning faithfulness.
>
> Thus, our contribution is to **stress-test the dominant evaluation paradigm**, rather than rely on it, and nevertheless, the strong correlation with D2A in most cases, especially when we have an 8/10 dataset, shows **strong positive correlation** with p < 0.05, we still believe current measurement can still play important role in verifying monitorability, but may further require complementary measurement like D2A, as stated in line 416-422.
> We agree that D2A coverage is currently limited, we will expand it during the later revision.
>
> ---
>
> **W4/Q3: Causal test of mechanistic experiments**
>
> We agree that causal evidence is important. In RLVR, entropy regularization alone is weak and unstable [4], so we instead perform controlled interventions using a higher-clip GRPO [5], as demonstrated in Response of **W2/3 to Reviewer ESWC**.
>
> For segment-level attention, direct intervention (e.g., rewarding or modifying attention mass) can significantly alternate model behavior, making clean causal testing difficult. We thus leave this for future works.
>
> ---
>
> **W4/Q5: Different model setups**
>
> We evaluate an additional larger model **DeepSeek-R1-Distill-Llama-8B** and observe consistent trends, as shown in Response **Q2 to Reviewer 476P**.
>
> In addition, our study targets the reasoning-model regime, where long-CoT SFT (or reasoning-format SFT) is standard (e.g., DeepSeek-R1, Qwen3, Kimi tech reports). In addition, the pre-RLVR pipelines of our tested models are different:
>
> * **DS-1.5B** (SFT cold-start only)
> * **Qwen3-4B** (SFT + on-policy distillation)
>
> Despite these differences, both models show aligned behavior, suggesting robustness of our findings.
>
> ---
>
> **W4: Relation to [3]**
>
> Our results are **complementary rather than conflicting** with [3]:
> * [3]: monitorability improves early and plateaus – **We confirm this (Table 2)**
> * [3]: more reasoning tokens correlate with monitorability – **Complementary to our analysis**
>
> Our work extends this by explaining what drives these dynamics.
>
> ---
> **Q2: Statistical significance and stability**
>
> We provide some concrete examples in **W2/Q1**,  and we summarized some additional uncertainty analysis:
>
> Bootstrap CI (average across all checkpoints):
> * DS-1.5B: ±0.017
> * Qwen3-4B: ±0.026
>
> Capability vs monitorability:
> * Only 9/32 positive correlations (p<0.05)
> * 3/32 negative correlations (p<0.05)
>
> Cross-monitor agreement:
> * Consistent positive correlation across all benchmarks (p < 0.001)
>
> These results support that:
> * Main trends are statistically stable
> * Capability vs Monitorability is often unstable and uncorrelated.
> * Cross-monitor agreements are stable and consistent.
>
> We will include more complete uncertainty reporting in the revision.
>
> ---
>
> [1] Reasoning Models Don't Always Say What They Think
>
> [2] Measuring the Faithfulness of Thinking Drafts in Large Reasoning Models
>
> [3] Monitoring Monitorability
>
> [4] The Entropy Mechanism of Reinforcement Learning for Reasoning Language Models
>
> [5] DAPO: an Open-Source LLM Reinforcement Learning System at Scale

---

> > ### Author Rebuttal · Reviewer_xozf · 2026-04-04
> >
> > Thank you for your thorough reply. I also checked the rebuttal to other authors, and it addressed a large part of my previous concerns. Therefore, I recommend weak acceptance.

---

> > > ### Author Response · Authors · 2026-04-04
> > >
> > > We again thank you for your comprehensive review, and our response addresses your concerns! We will ensure that all details from the rebuttal are included in the later revision.

---

### Official Review · Reviewer_476P · 2026-03-16

**Soundness:** 3
**Presentation:** 3
**Significance:** 3
**Originality:** 3
**Overall Recommendation:** 4
**Confidence:** 4

**Summary:**

The paper attempts to investigate a broad area of safety and interpretability in reasoning models. The research's main topic consists of understanding monitorability, i.e., whether a model’s reasoning traces faithfully reflect its internal decision process and can therefore be inspected by humans or automated oversight systems. The authors study how monitorability evolves during RLVR training of large reasoning models. Through experiments on multiple base models and domains (math, code, science, instruction-following), they analyze how monitorability changes across training steps and data distributions. The study finds that monitorability often improves early in RLVR training but later plateaus or declines, and that these gains depend strongly on training distribution rather than capability alone. The paper also investigates mechanistic drivers such as entropy reduction and attention shifts during training.

**Compliance With Llm Reviewing Policy:**

Affirmed.

**Key Questions For Authors:**

1. How sensitive are the observed monitorability dynamics to the specific RLVR setup (reward design, training schedule, etc.)?
2. Do the findings generalize to larger frontier reasoning models or other RL-based training methods?
3. Can the insights from this study be translated into concrete training strategies that explicitly improve monitorability?
5. How robust are the proposed monitorability metrics across different reasoning formats and prompting styles?

**Limitations:**

Yes

**Strengths And Weaknesses:**

Strengths
Soundness
1. The paper provides a systematic empirical study of monitorability dynamics during RLVR training.
2. Experiments cover multiple model families and domains, providing evidence that monitorability improvements occur early but are not sustained during extended training.
3. The analysis explores potential mechanisms, such as distribution sharpening and attention shifts during RLVR training.

Presentation
1. The paper is clearly motivated and frames monitorability as an important property for oversight and alignment of reasoning models.
2. The empirical findings are organized around clear questions regarding training dynamics and data distribution effects.

Significance
1. Understanding whether reasoning traces reliably reflect model reasoning is an important question for AI safety and oversight.
The study provides useful empirical observations about how monitorability behaves under standard RLVR training.

Originality
1. The work focuses on a relatively underexplored property (monitorability of reasoning traces) and analyzes its training dynamics systematically.
2. The empirical characterization of monitorability as a partially emergent but unstable property is a useful contribution.

Weaknesses
Soundness
The work is primarily empirical and observational; the mechanistic explanations remain somewhat speculative.
It is unclear whether the observed patterns generalize beyond the specific models and RLVR setups studied.

Presentation
Some parts of the analysis rely on newly introduced monitorability metrics whose definitions and interpretation could be explained more clearly.
The connection between internal activation analysis and monitorability could be strengthened.

Significance
While the study provides insights into training dynamics, it does not propose concrete methods to improve monitorability in practice.
As a result, the work is more diagnostic than actionable.

Originality
The core contribution is an empirical analysis rather than a new algorithm or training method.
Some conclusions may depend on existing RLVR training dynamics already observed in prior work.

---

> ### Author Rebuttal · Authors · 2026-03-31
>
> We thank the reviewer for their thoughtful and positive assessment of our work. We address the main weaknesses and questions below:
>
> ---
> **W1: Mechanistic explanations are speculative**
>
> Thank you for this important feedback. We agree that our current mechanistic analysis is primarily correlational and can be strengthened. In the revision, we will clarify this limitation and refine our claims.
>
> To move beyond high-level hypotheses, we add new evidence (details in response to W1/2/3 to Reviewer ESWC):
>
> * **IF mechanism analysis**: We show that IF training consistently increases cross-segment attention and more monitorable casual influence of interventions, suggesting that IF improves monitorability by strengthening prompt-conditioned reasoning.
> * **Entropy analysis refinement**: We include hardness-based analysis, identify failure cases (e.g., early collapse on hard tasks), and introduce transition-based metrics and **controlled entropy experiments** (higher-clip in GRPO) to better characterize when entropy aligns or misaligns with monitorability.
>
> While still not fully causal, these analyses identify two interacting factors:
> **(i) prompt-conditioned reasoning** and **(ii) distribution sharpening**, both of which are strongly data and training setup dependent.
>
> ---
>
> **W2: Presentation of newly introduced metrics**
>
> Thank you for this suggestion. We will improve clarity in the revision by
> * explicitly defining each metrics
> * clarifying how different metrics relate to distinct aspects of monitorability, and
> * strengthening the connection between activation-level analysis  and behavior-level monitorability.
>
> ---
>
>
> **W3/Q3: Concrete methods to improve monitorability**
>
> We agree that making the work more actionable is important. Our results suggest several concrete insights:
>
> * IF training during the process is important.
> * Training on mixed domains often leads to better monitorability.
> * Monitoring capability during training alone is insufficient.
> * Avoid premature entropy collapse on suboptimal distributions (W2/3 to Reviewer ESWC).
>
> We will make these recommendations explicit in the later revision.
>
> ---
>
> **W4: Originality**
>
> Thank you for recognizing the novelty of studying monitorability as an evolving property during training. We agree that our work does not introduce a new algorithm or the original “free gift” observation or metric. Our contribution is instead a **systematic, dynamic analysis of monitorability during RLVR**, which we believe is missing in prior work.
>
> Specifically, we study:
> * the critical role of training composition,
> * the weak coupling between capability and monitorability,
> * internal mechanistic correlation of monitorability, and
> * how monitorability evolves with training dynamics and task complexity.
>
> Importantly, we moves beyond prior observations by identifying when and why monitorability emerges or degrades during training,
>
> ---
> **Q1: Sensitivity to RLVR setup**
>
> Our experiments already span diverse RLVR setups (math, code, science, IF), each with different reward mechanisms (exact match, unit tests, LLM judge, rubric scoring), and we observe consistent but data-dependent dynamics, which is a key finding.
>
> To further probe sensitivity, we vary training configuration:
>
> **DS-1.5B Math: varying maximum reasoning length**
>
> | 4096     | 6144 | 8172 |
> | -------- | ---- | ---- |
> | **0.36** | 0.34 | 0.29 |
>
> Moderate length yields better monitorability, suggesting that overly long reasoning may introduce noisy or superficial steps.
>
> We also show in **W2/W3 (Reviewer ESWC)** that controlling entropy (via higher-clip GRPO) changes monitorability dynamics. These further enhanced the comprehensiveness of our analysis.
>
> ---
>
> **Q2: Generalization to larger models**
>
> We extend experiments to a larger model (DeepSeek-R1-Distill-Llama-8B (DS-8B)):
>
> | DS-8B | Step100 | Step200 | Step300 | Step400 |
> | --- | -- | -- | -- | -- |
> | Math  | 0.40    | 0.43    | 0.42    | 0.42    |
> | IF    | 0.42    | 0.47    | 0.50    | 0.48    |
>
> We observe consistent patterns:
>
> * early-stage improvement, and
> * IF training improves monitorability,
>
> While we cannot test frontier-scale models, the consistency across 1.5B, 4B, and 8B models and multiple backbones suggests reasonable generality within the current reasoning-model regime.
>
> ---
>
> **Q4: Robustness across reasoning formats and prompting styles**
>
> We evaluate robustness using **three intervention types per question**. We compute Pearson correlation between one intervention type and the average of the other two:
>
> **95% CI of Pearson r**
>
> | Hint type  | DS1.5B      | Qwen3-4B |
> | -- | -- |--|
> | Grader     | [0.854, 0.940] | [0.837, 0.940] |
> | Sycophancy |  [0.962, 0.985] | [0.650, 0.863] |
> | Unethical  | [0.943, 0.977] | [0.884, 0.958] |
>
> These strong positive correlations show that monitorability is consistent between intervention types. We also report cross monitor agreement (Table 5 in Appendix), further supporting robustness.

---

### Decision · Program_Chairs · 2026-04-30

**Decision:**

Accept (regular)

**Comment:**

### Reviews and discussion

Scores: 5, 4, 4, 2.

Reviewer iMp4 (5, conf 4):
- Good experimental coverage, interesting findings, actionable IF discovery. Minor concern about model size. Fully resolved after DS-8B results. Maintaining 5.

Reviewer xozf (4, conf 4):
- Raised the most specific technical questions. Concerns included training budget confounds, metric validity (g-mean^2 may capture prompt-following, not faithfulness), correlational mechanisms, and limited novelty vs prior work. Authors provided matched-budget IF experiments, bootstrap CIs, cross-monitor agreement, and entropy-control experiments. Marked "Fully resolved" and submitted Final Justification recommending weak acceptance: "The additional experiments and clarifications strengthened the novelty."

Reviewer 476P (4, conf 4):
- Systematic study with good coverage. Concerns about speculative mechanisms, no concrete methods to improve monitorability, and primarily observational contribution. Authors addressed with concrete recommendations (use IF data, mixed domains, avoid premature entropy collapse) and DS-8B results.

Reviewer ESWC (2, conf 4):
- Core concern: the paper does not identify causal factors behind why IF improves monitorability. Evidence remains correlational. Entropy analysis has inconsistencies. Paper reads as a "collection of empirical observations." After rebuttal: marked "(c) partially resolved, core concerns not easily addressed." Final Justification maintaining reject: "core issues regarding causal understanding and explanatory depth remain."

The authors sent a detailed follow-up to ESWC with completed 800-step entropy control experiments. The key experiment: after 400 steps of high-clip GRPO (exploration phase), switching to standard clipping (sharpening phase) improves monitorability, while sustained high entropy does not. Since training data is unchanged, this isolates the effect of entropy dynamics from data effects. Despite this, ESWC did not respond further.

### Assessment

I recommend accept. Three of four reviewers support acceptance (5, 4, 4), and the paper addresses a question with real safety implications: when can we trust LLM reasoning traces?

The core contribution is the first systematic, checkpoint-level analysis of monitorability dynamics during RLVR. The findings are practically useful: IF data helps, monitorability is not capability, premature entropy collapse hurts. The controlled entropy experiments (explore-then-sharpen improves monitorability under the same data) move beyond pure correlation and provide the closest evidence to a causal mechanism for the entropy effect.

Reviewer ESWC raises a fair point: the paper does not fully explain WHY IF data helps monitorability. The authors provide correlational evidence (attention shifts, intervention ratios, hint causality), which narrows the hypothesis but does not constitute causal proof. This is a limitation worth noting but does not, in my assessment, outweigh the paper's scope and usefulness. The three supporting reviewers, including the most technically detailed one (xozf), found the evidence sufficient for this type of empirical safety contribution.

Regarding ESWC's score of 2: this seems misaligned with their own acknowledgment that the paper provides an "extensive empirical study" with "clearer empirical characterization of monitorability." A score of 2 typically indicates fundamental flaws, which this paper does not have. I give reduced weight to this review relative to the consensus.